# Formation of Myoglobin Corona at Polymer Microparticles

Zbigniew Adamczyk * and Małgorzata Nattich-Rak 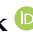

Jerzy Haber Institute of Catalysis and Surface Chemistry Polish Academy of Science, Niezapominajek 8 Street, 30-239 Cracow, Poland; ncnattic@cyf-kr.edu.pl
* Correspondence: ncadamcz@cyf-kr.edu.pl; Tel.: +48-126-395-104; Fax: +48-124-251-923

**Abstract:** Adsorption of myoglobin molecules at negatively charged polystyrene microparticles was studied using the dynamic light scattering (DLS), electrophoresis (LDV) and the solution depletion method involving atomic force microscopy (AFM). The measurements were carried out at pH 3.5 and NaCl concentration of $10^{-2}$ and 0.15 M. Initially, the stability of myoglobin solutions and the particle suspensions as a function of pH were determined. Afterward, the formation of myoglobin molecule corona was investigated via the direct electrophoretic mobility measurements, which were converted to the zeta potential. The experimental results were quantitatively interpreted in terms of the general electrokinetic model. This approach yielded the myoglobin corona coverage under in situ conditions. The maximum hard corona coverage was determined using the AFM concentration depletion method. It was equal to 0.9 mg m$^{-2}$ for the NaCl concentration in the range 0.01 to 0.15 M and pH 3.5. The electrokinetic properties of the corona were investigated using the electrophoretic mobility measurements for a broad pH range. The obtained results confirmed that thorough physicochemical characteristics of myoglobin molecules can be acquired using nM amounts of the protein. It was also argued that this method can be used for performing electrokinetic characteristics of other proteins such as the SARS-Cov-2 spike protein exhibiting, analogously to myoglobin, a positive charge at acidic pHs.

**Keywords:** adsorption of myoglobin; myoglobin corona at microparticles; stability of myoglobin corona; zeta potential of myoglobin corona; zeta potential of myoglobin

## 1. Introduction

The formation of protein coronas at nanoparticles was extensively studied, both for single molecule systems and for mixtures comprising the blood serum [1–6]. It is generally assumed that there exists a fraction of irreversibly bound protein forming the hard corona and a fraction of less tightly bound molecules forming the soft corona exhibiting a fuzzy, gel like structure. Because of relatively low stability, soft corona composition and structure are difficult to be determined by available experimental techniques, which require nanoparticle suspension centrifugation or filtration steps. Another disadvantage consists in the destabilization of the suspension by the soft coronas, which may promote bringing interactions among nanoparticles. Last not least, the presence of soft corona hinders quantitative investigations of the hard coronas, whose density and structure could be otherwise predicted in an adequate way applying theoretical modeling.

Therefore, one can argue that more reliable results prone to a quantitative interpretation can be acquired if the formation of soft corona is prohibited. Additionally, the reliability of corona formation measurements can be significantly increased using particles of larger size, which yields an adequate stability of such systems. Experimental results acquired for albumins [7], immunoglobulin [8] and lysozyme [9] using polymer microparticles confirm that these systems are prone to a theoretical interpretation.

In this work we focus attention on myoglobin whose main function consists in oxygen storage and transport in tissues [10,11] as well as the control of the nitric oxide flux in the heart under physiological and pathological conditions [12]. Myoglobin can also

serve as a marker of myocardial infraction and acute renal failure [13]. The molar mass of myoglobin molecule derived from chemical composition is equal to 17,800 g mol$^{-1}$, which was confirmed by neutron scattering measurements [14] and its density is equal to 1.35 g cm$^{-3}$ [15]. Electrokinetic properties of the molecule were studied in Reference [16], where it is shown that molecule exhibits quite analogous charge distribution at acidic pHs to the receptor part of the spike protein of the SARS-Cov-2 virus.

Adsorption of myoglobin on zirconium phosphate, zirconium benzenephosphonate and phosphate grafted zirconia nanoparticles was studied in Refs. [17,18] for the pH range 5–8. In Reference [19], the role of ionic strength and pH in myoglobin adsorption on ordered mesoporous silicates was determined, whereas adsorption isotherms and conformational changes of myoglobin upon adsorption on hydroxyapatite were studied by Iafisco et al. [20]. The catalytic activity in an oxidation process of myoglobin layers adsorbed on nanosized hydrotalcites were investigated in [21]. In Reference [22], the adsorption isotherms of myoglobin on silica nanoparticles of the size equal to 28 nm at pH 7.4 (0.1 M phosphate) were determined by the concentration depletion method. It is determined that the obtained isotherm could be fitted by the Langmuir model with the rather low maximum coverage of 0.25 mg m$^{-2}$.

Thorough investigations of myoglobin corona formation at silica nanoparticles of the size 50 nm at pH the pH range 4–10 and different ionic strength were performed in Reference [23] also applying the concentration depletion method. The amount of protein molecules forming the hard and soft corona as well as their equilibrium adsorption constant were determined. It is shown that for the 0.1 M NaCl concentration the myoglobin coverage in the hard corona was equal to 0.9 mg m$^{-2}$ at pH 4 and y decreased to 0.5 mg m$^{-2}$ at pH 10. These effects were interpreted as due to the electrostatic interactions controlled by the surface charge regulation effect.

However, in the above mentioned works the electrokinetic properties of the myoglobin/nanoparticle complex were not studied because of a limited nanoparticle suspension stability especially at larger pHs.

Given the deficit of reliable information, the goal of this work is to determine the mechanism of myoglobin adsorption at polymer microparticles with the emphasis focused on the hard corona formation. The adsorption is monitored under in situ conditions via the electrophoretic mobility measurements, which are converted to the zeta potential. These results are interpreted in terms of a general electrokinetic model enabling the myoglobin coverage at microparticles to be determined even for the low concentration range. Additionally, the maximum coverage of the hard corona is determined using the concentration depletion method involving the atomic force microscopy (AFM). The stability and electrokinetic properties of the hard corona for a broad pH range are also investigated.

It is expected that that exploiting these experimental data one can develop a robust procedure for preparing stable myoglobin coronas characterized by well-defined coverage, which can be used for efficiently performing various immunological assays.

## 2. Materials and Methods

Myoglobin stemming from the equine hart supplied in the form of a lyophilized powder 95–100% ((Sigma-Aldrich) St. Louis, MO, USA) was used in this work.

Suspension of negatively charged polystyrene particles bearing sulfate surface groups were a commercial products of Invitrogen. The stock suspension was purified by a thorough membrane filtration and afterward diluted in the myoglobin adsorption experiments to a desired mass concentration, typically 50 mg L$^{-1}$.

Natural ruby mica (Continental Trade, Warsaw, Poland) was used for as a solid substrate for the residual myoglobin adsorption studied by atomic force microscopy (AFM) imaging. Thin sheets of mica were freshly cleaved before each experiment and used without any pretreatment.

Water was purified using a Milli-Q Elix and Simplicity apparatus. The sodium chloride, hydrochloric acid and sodium hydroxide were commercial products of Sigma-Aldrich and used without additional purification.

The effective bulk concentration of myoglobin after dissolving the powder in appropriate electrolyte and pH and after filtration, was spectrophotometrically determined using a Shimadzu UV-2600 apparatus exploiting the peak absorption at 409 nm.

The diffusion coefficients of myoglobin molecules and polymer particles were determined by dynamic light scattering (DLS) using the Zetasizer Nano ZS instrument (Malvern, Cambridge, UK). The hydrodynamic diameter was calculated from the Stokes-Einstein equation. The polymer particle size distribution was also determined using the laser diffractometer by the Particle Size Analyzer LS 13,320 (Beckman-Coulter).

The electrophoretic mobility of myoglobin molecules and the particles were measured using the Laser Doppler Velocimetry (LDV) technique using the same Zetasizer Nano ZS instrument from Malvern (Cambridge, UK). The zeta potential was calculated using the the Henry and Smoluchowski formulae, respectively.

In the DLS and the LDV measurements myoglobin solutions of the concentration equal to 300–500 mg L$^{-1}$ were used whereas in the adsorption experiments they were diluted by a sodium chloride solution to the desired bulk concentration in the range 0.1 to 1 mg L$^{-1}$. The pH in the range of 3 to 5 was adjusted by the addition of HCl, the pH of 7.4 was fixed by the PBS buffer and larger pHs were adjusted by NaOH.

The temperature of experiments was kept at a constant value equal to $298 \pm 0.1$K.

Atomic force microscopy (AFM Moscow, Moscow, Russia) measurements were carried out using the NT-MDT OLYMPUS IX71 device with the SMENA scanning head. The measurements were performed in semi-contact mode using silicon probes (polysilicon cantilevers HA-NC ETALON with resonance frequencies of 140 kHz +/− 10% or 235 kHz +/− 10%.

## 3. Results and Discussion

### 3.1. Physicochemical Characteristics of Myoglobin Molecules and Polymer Particles

Initially, the stability of myoglobin solutions under various physicochemical conditions was investigated. For this purpose, the diffusion coefficient was measured by DLS as a function of the storage time. It is confirmed the molecule diffusion coefficient was practically independent of pH within the range 3–8 and assumed an average value of $1.2 \pm 0.1 \times 10^{-6}$ cm$^2$ s$^{-1}$. This corresponds to the hydrodynamic diameter of the myoglobin molecules $d_H$ equal to $4.1 \pm 0.2$ nm (calculated from the Stokes–Einstein relationship). After 400 min it slightly increased to $4.2 \pm 0.1$ nm, (see Figure 1), which suggests that the myoglobin solutions were stable for the above time period. A more significant changes in the hydrodynamic diameter only occurred after the time of 20 h.

In an analogous way, applying the DLS measurements, the hydrodynamic diameter of the polymer particles was determined. It was equal to $830 \pm 20$ and $820 \pm 30$ nm for NaCl concentration equal to 0.01 and 0.15 M, NaCl, respectively. The particle size distribution was also determined using the laser diffractometry measurements and is shown in Figure 2. The average particle size acquired in this way was equal to $810 \pm 30$ nm a for NaCl concentration equal to 0.01 M. It should be mentioned that the particle suspensions were stable for both NaCl concentrations and pH 3–8 for the storage time of 48 h.

The electrophoretic mobility of myoglobin molecules and microparticles was measured using LDV for NaCl concentration equal to 0.01 and 0.15 M. Using these electrophoretic mobility data, the zeta potential was calculated. The results shown in Figure 3 indicate that the myoglobin zeta potential is positive for H below 5 attaining 38 and 15 mV at pH 3.5, for NaCl concentration of 0.01 and 0.15 M, respectively. At pH > 5, the zeta potential becomes negative. However, its absolute values are rather small, especially for the 0.15 M NaCl concentration, prohibiting a precise determination of the myoglobin isoelectric point (iep).

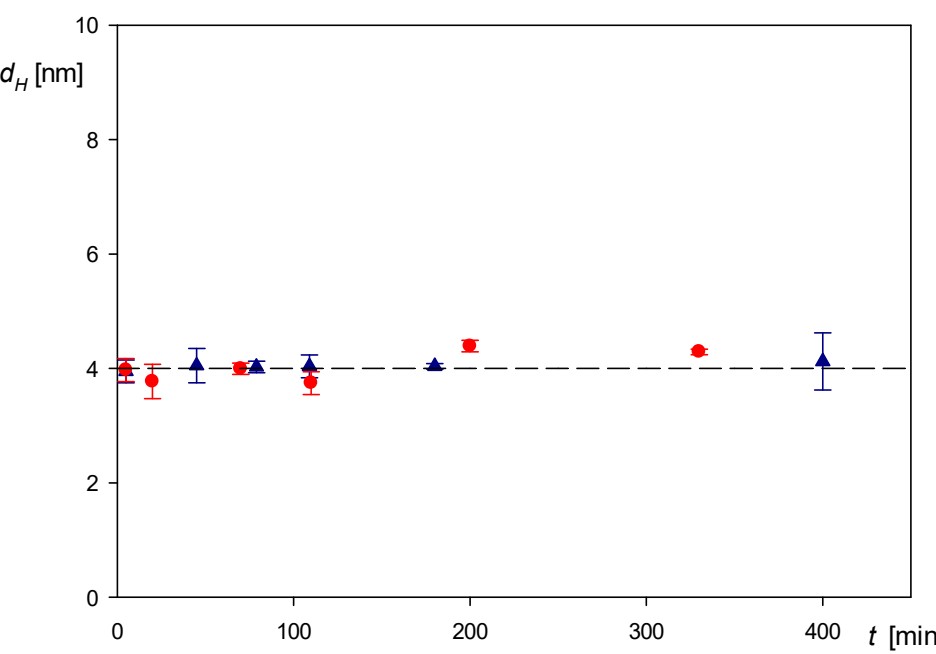

**Figure 1.** Dependence of the myoglobin hydrodynamic diameter on the storage time derived applying DLS measurements for NaCl concentration of 0.01 M; 1. ● pH 3.5, 2. ▲ pH 7.4 (PBS), and 0.15 M. The dashed line shows the initial value of the hydrodynamic diameter equal to 4.1 nm.

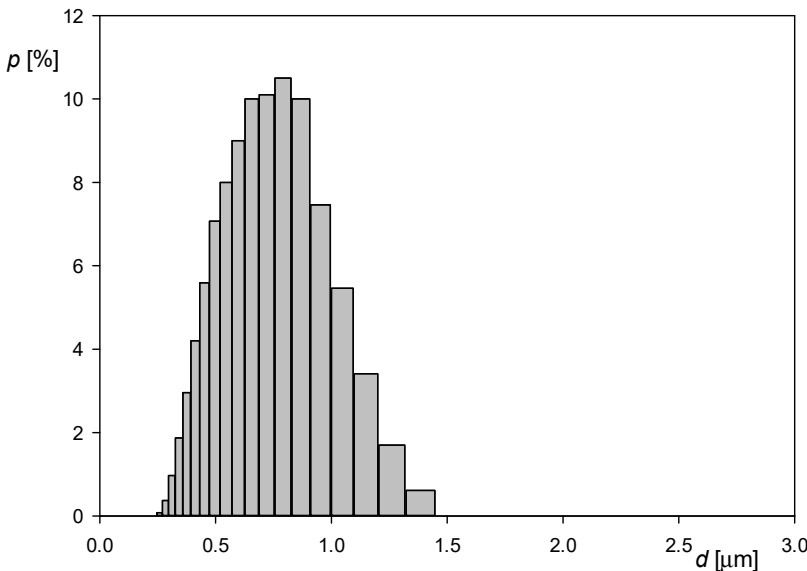

**Figure 2.** Histogram of the microparticle size distribution derived from laser diffractometry, the average particle size is equal to $810 \pm 30$ nm.

On the other hand, the zeta potential of particles was markedly negative for the entire pH range and equal to $-85$ and $-60$ mV for the NaCl concentration of $10^{-2}$ M and 0.15 M, respectively (see Figure 2).

### 3.2. Formation of Myoglobin Corona

The procedure of the myoglobin corona formation on polymer particles was analogous to that described previously in References [7–9]. Initially, the protein solution of the concentration varied between 0.1 and 5 mg L$^{-1}$ was mixed with the suspension of microparticles of a controlled bulk concentration typically equal to 50 mg L$^{-1}$ and incubated over 15 min at room temperature. Afterward, the electrophoretic mobility of protein covered particles

was measured. In this way, the primary dependencies of the electrophoretic mobility of microparticles on the amount of added protein were acquired. It should be underlined that this procedure was reproducible, enabling to derive in a reliable way dependencies of the electrophoretic mobility and the zeta potential of particles on the initial protein concentration in the suspension.

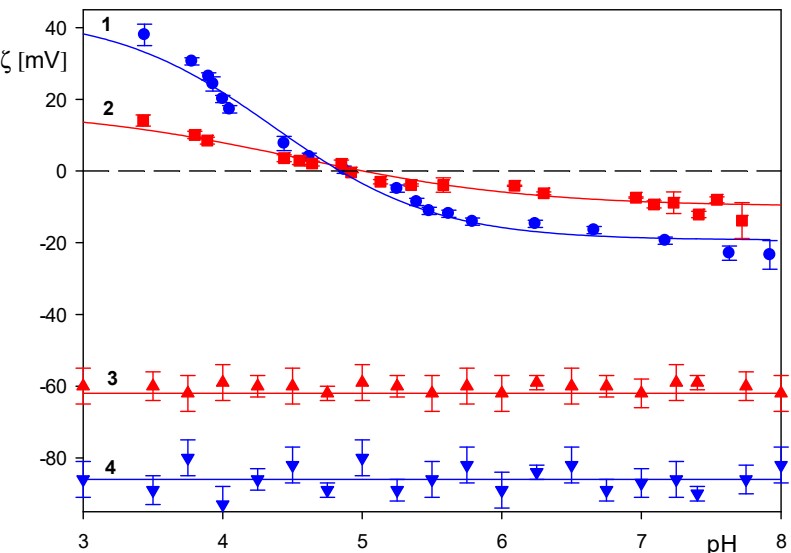

**Figure 3.** Dependence of the zeta potential on pH derived from the LDV measurements: (1) myoglobin in 0.01 M, NaCl; (2) myoglobin in 0.15 M, NaCl; (3) microparticles in 0.15 M, NaCl; (4) microparticles in 0.01 M, NaCl. The solid lines are non-linear fits of experimental data.

It is also worth mentioning that the kinetics of corona formation at polymer microparticles is significantly faster compared to the adsorption at planar substrates [24]. Moreover, it is practically independent of the bulk protein concentration, whereas for planar substrates the monolayer formation time increases inversely proportionally to the bulk protein concentration, exceeding many hours for the bulk concentration below 1 mg L$^{-1}$. The corona formation time $t_c$ can be calculated from the following formula [24]

$$t_c = \left( \frac{\Phi_{mx}\rho_m}{c_m} \right)^{\frac{2}{3}} \frac{d_m^2}{4\bar{D}} \qquad (1)$$

where $\Phi_{mx}$ is the maximum volume fraction of polymer particles in the suspension that is equal to 0.62 for a random configuration of spheres, $\rho_m$ and $d_m$ are the particle density and the diameter, respectively, $c_m$ is the particle bulk concentration and $\bar{D}$ is the effective diffusion coefficient, which is the sum of the protein molecule and the particle diffusion coefficients.

Using the parameters pertinent to our measurements, i.e., $c_m$ = 50 mgL$^{-1}$, $\rho_m$ = 1.05 $\times$ 10$^3$ kg m$^{-3}$, $d_m$ = 8.3 $\times$ 10$^{-7}$ m, and $\bar{D}$ = 1.2 $\times$ 10$^{-10}$ m$^2$ s$^{-1}$ one obtains from Equation (1) that $t_c$ = 3.2 s, which is considerable shorter than the experimental incubation time, equal to 900 s.

In Figure 4 the corona formation results are shown as the dependence of the zeta potential of particles (calculated from using the Smoluchowski equation) on the initial myoglobin concentration in the suspension $c_b$. As can be seen, the zeta potential rapidly increases and becomes positive for the myoglobin concentration larger than ca. 0.2 mg L$^{-1}$. It is also worth underlining that for 0.01 M NaCl, the initial slope of the zeta potential vs. the bulk protein concentration exceeds 500 mV L mg$^{-1}$. Considering that the zeta potential can be determined with a precision of 1 mV, one can predict that applying the

LDV technique one can detect in a robust way myoglobin concentration of the order of 1 nM. However, for bulk protein concentration larger than 0.2 mg L$^{-1}$ changes in the zeta potential become rather moderate. Finally plateau values of zeta potential are attained equal to 20 and 5 mV for 0.01 and 0.15 M, NaCl, respectively. They are markedly smaller than the limiting bulk myoglobin zeta potential values equal to 38 and 15 mV for the 0.01 and 0.15 M, NaCl concentration, respectively.

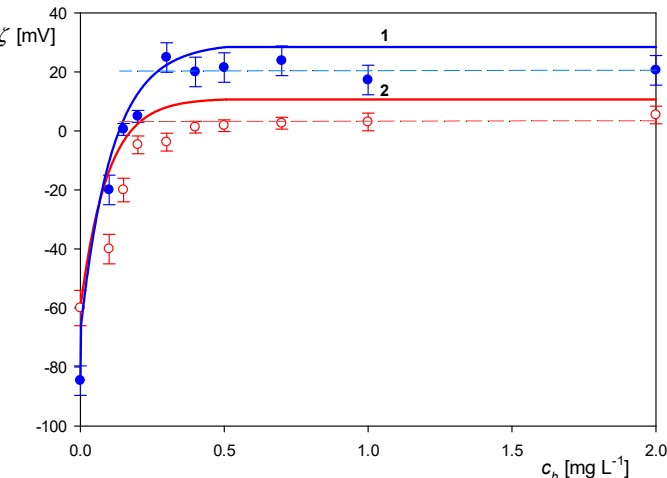

**Figure 4.** Dependence of the zeta potential of the particles on the bulk myoglobin concentration: (1). ● 0.01 M, NaCl, (2). ○ 0.15 M, pH 3.5. The points denote experimental data derived from the LDV measurements and the solid lines show theoretical results calculated using the general electrokinetic model.

The primary experimental data shown in Figure 4 were interpreted in terms of the theoretical electrokinetic model formulated in Reference [25]. In contrast to the Gouy-Chapmann approach, in this model three-dimensional fluid velocity and electric potential distributions around adsorbed protein molecules are considered in an exact way applying the multiple expansion method. This enabled the following expression for the zeta potential of interfaces covered by protein molecules $\zeta(\Theta)$ to be formulated.

$$\zeta(\Theta) = F_i(\Theta)\zeta_i + F_p(\Theta)\zeta_p \tag{2}$$

where $\Theta$ is the absolute (dimensionless) coverage of protein molecules, $\zeta_i$ is the zeta potential of bare substrate, $\zeta_p$ is the particle (protein) zeta potential in the bulk, and $F_i(\Theta)$, $F_p(\Theta)$ are the dimensionless functions. The $F_i$ function describes the damping of the flow within the adsorbed molecule layer and the $F_p$ function characterizes the contribution to the zeta potential stemming from the molecules. Accordingly, for low particle coverage, the $F_i$ function approaches unity and the $F_p$ function vanishes. For thin double-layers, one can express the functions by the following analytical expressions [25]

$$F_i(\Theta) = e^{-C_i\Theta}$$
$$F_p(\Theta) = a_p\Theta + b_p\left(1 - e^{-C_i\Theta}\right) \tag{3}$$

where the $C_i$, $a_p$ and $b_p$ coefficients for spherical particles layers assume the limiting values of 10.2, 0.202 and 0.618, respectively.

The absolute coverage in Equations (2) and (3) can be calculated as:

$$\Theta = S_g N = S_g\left(\frac{Av}{M_w}\right)\Gamma_c \tag{4}$$

where $S_g$ is the characteristic cross-section area of the protein molecule, $Av$ is the Avogadro constant, $M_w$ is the molar mass of the protein and $\Gamma_c$ is the nominal coverage of the protein corona connected with the bulk protein concentration by the dependence:

$$\Gamma_c = \left( \frac{\rho_m d_m}{6} \right) \frac{c_b}{c_m} \qquad (5)$$

The results calculated from Equations (3)–(5) are plotted as solid lines in Figure 4. As can be seen, they adequately reflect the main features of the experimental runs especially for 0.01 M NaCl albeit for the bulk myoglobin concentration below 0.3 mg L$^{-1}$. This confirms an irreversible adsorption of myoglobin for this bulk suspension concentration range. A more precise estimate of the maximum coverage can be acquired as the intersection of the horizontal lines approximating the zeta potential for large $c_b$ with the theoretical lines derived from Equation (3). In this way one obtains that the limiting concentration was equal to 0.27 mg L$^{-1}$ (for the NaCl concentration of 0.01 M) that corresponds according to Equation (5) to the maximum myoglobin corona coverage of 0.80 ± 0.2 mg m$^{-2}$. For 0.15 M NaCl, one can estimate obtains in an analogous way that the maximum coverage was equal 0.75 ± 0.2 mg m$^{-2}$. These results agree with the coverage of myoglobin hard corona on silica particles previously determined by Lee et al. [23] at pH 4 and with the maximum coverage of lysozyme on polymer microparticles [9] which was equal to 0.8 and 0.9 mg m$^{-2}$ for 0.01 M and 0.15 M NaCl, respectively (at pH 3.5). However, it should be mentioned that the precision of the maximum coverage determination via the LDV method does not exceed 0.1 mg m$^{-2}$ because it relies on small variations of the zeta potential appearing for larger bulk protein concentration.

In order to increase the precision of the maximum coverage determination a concentration depletion method exploiting AFM was applied, which yields the residual concentration of myoglobin in the solution remaining after the adsorption on microparticles. According to this procedure previously applied for albumins [7] and lysozyme [9] the particle/myoglobin mixture acquired after the corona formation step is transferred to a thermostated cell. Then, a few freshly cleaved mica sheets are immersed into the mixture and the residual myoglobin molecule are allowed to adsorb under diffusion-controlled transport over the time typically equal to 30 min. It should be mentioned that the particle deposition is negligible during this time because of their small number concentration and much smaller diffusion coefficient. Afterward, the mica sheets covered by the protein are rinsed and dried. The average number of adsorbed myoglobin molecules $N_p$ is determined over a few equal sized spots randomly chosen over the mica sheets using AFM imaging. Knowing $N_p$, the surface concentration of myoglobin on mica was calculated as $N_m = N_p/S_l$ (where $S_l$ is the surface area of one spot). Under the diffusion transport, the myoglobin surface concentration is connected with the residual concentration in the suspension after the corona formation step $c_r$ by the formula [26]

$$N_m = C_m c_r = C_m (c_b - c_{ads}) \qquad (6)$$

where $C_m = 2(Dt_a/\pi)^{1/2}$, $t_a$ is he adsorption time, and $c_{ads}$ is the concentration of the adsorbed (depleted) myoglobin forming the corona.

One can infer from Equation (6) that for $N_m = 0$ the amount of protein adsorbed $c_{ads}$ is equal to the initial protein concentration in the solution equal to $c_b$.

The results of AFM measurements carried out at pH 3.5 and NaCl concentration of $10^{-2}$ M plotted as the dependence of on $N_m$ on $c_b$ are shown in Figure 5. One can observe that for $c_b$ up to 0.3 mg L$^{-1}$ the myoglobin adsorption on mica was negligible. This means that the entire myoglobin concentration initially present in the solution was consumed for the corona formation. For $c_b$ larger than 0.3 mg L$^{-1}$ an abrupt and linear increase in $N_m$ was observed. Hence, these measurements indicate that the maximum coverage of the hard corona calculated from Equation (5) using the above threshold $c_b$ concentration of 0.3 mg L$^{-1}$ is equal to 0.87 ± 0.1 mg m$^{-2}$. In an analogous way, for 0.15 M, NaCl one

obtained $0.90 \pm 0.1$ mg m$^{-2}$. These measurements confirmed that one can produce hard coronas on polymer particles of well-controlled coverage attaining 0.9 mg m$^{-2}$.

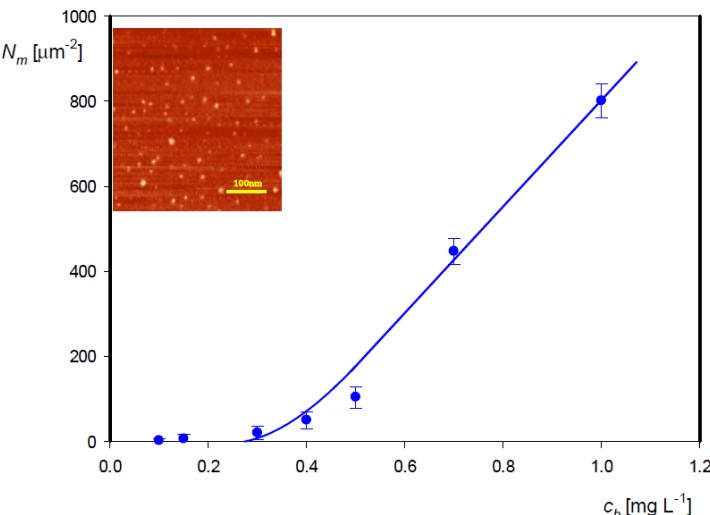

**Figure 5.** Dependence of the surface concentration of residual myoglobin molecules adsorbed on mica $N_m$ on the initial myoglobin concentration $c_b$ in the suspension for 0.01 M, NaCl, pH 3.5. The points denote experimental data obtained from AFM measurements and the solid line shows is a non-linear fit of experimental data. The inset shows the myoglobin layer at mica imaged by AFM.

The stability of the polymer particles with adsorbed coronas was determined measuring their diffusion coefficient and electrophoretic mobility as a function of storage time. These primary data were converted to the hydrodynamic diameter and the zeta potential, respectively and shown in Figure 6 as the dependence of the normalized zeta potential and the normalized hydrodynamic diameter on the storage time. As can be seen, the particles bearing the corona characterized by the coverage of 0.87 mg m$^{-2}$ were stable over the time up to 500 min, which suggests that they are prone to long-lasting electrokinetic investigations. This finding is of a practical significance given that the corona formation only requires a myoglobin concentration of 0.3 mg L$^{-1}$, whereas in the bulk LDV measurement at least two orders of magnitude larger bulk concentrations are needed.

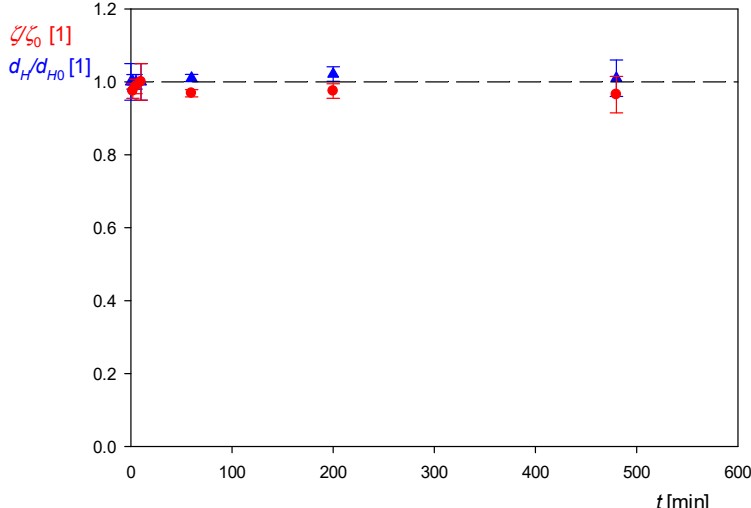

**Figure 6.** Stability of the polymer particles with the myoglobin corona of the coverage 0.87 mg m$^{-2}$ expressed as the dependence of the normalized zeta potential $\zeta/\zeta_0$ (▲), and the normalized hydrodynamic diameter $d_H/d_{H0}$ (●) on the storage time (where $\zeta_0$ and $d_{H0}$ are the zeta potential and the hydrodynamic diameter for the initial time); pH 3.5 and 0.01 M, NaCl.

The utility of hard coronas on polymer particles for performing robust electrokinetic characteristics of protein molecules is illustrated in Figure 7 where the pH dependence of the zeta potential of myoglobin molecules in the bulk is compared with analogous dependence obtained for the myoglobin corona characterized by the coverage of 0.85 mg m$^{-2}$.

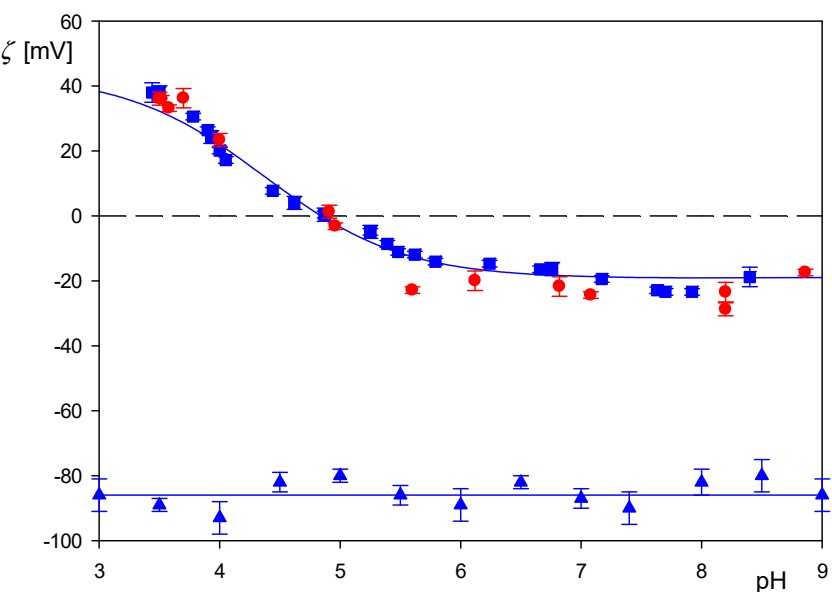

**Figure 7.** Dependence of the zeta potential on pH derived from the LDV measurements for 0.01 M, NaCl: ■, myoglobin molecules in the bulk, •, the myoglobin/particle corona (the coverage of 0.87 mg m$^{-2}$) transformed using Equation (7), ▲, bare polymer particles in the bulk. The lines are fits of experimental data.

The primary zeta potential values obtained for the corona were used to calculate the myoglobin zeta potential using the transformed version of Equation (2) in the following form:

$$\zeta_p = \zeta(\Theta)/F_p(\Theta) - F_i(\Theta)\zeta_i/F_p(\Theta) \tag{7}$$

where the absolute coverage was calculated using Equation (4) for the above corona coverage.

As one can see in Figure 7 the experimental data obtained for the corona transformed in this way agree within experimental error bounds with less effective bulk LDV measurements, requiring much larger amounts of the protein to be used.

## 4. Conclusions

The LDV electrophoretic mobility measurements supplemented by the AFM based concentration depletion methods enable a quantitative analysis of the myoglobin hard corona formation on polymer microparticles An adequate stability of the polymer particles with the hard corona characterized by the coverage up to 0.90 mg m$^{-2}$ was confirmed. This enabled to perform thorough measurements yielding the dependence of the corona zeta potential on pH. From these data one was able to extract using Equation (7) the zeta potential of the myoglobin molecules as a function of pH. The obtained results indicate that a thorough electrokinetic characteristics of myoglobin molecules can be acquired using nM amounts of the protein, whereas the conventional bulk characteristics require at least two orders of magnitude larger amounts. Hence, this procedure based on electrokinetic measurements has a particular significance for investigations of difficult to acquire proteins, for example the virus (SARS-Cov-2) spike protein.

Additionally, hard coronas characterized by well-defined coverage and electrokinetic properties can be exploited for quantitative studies of their interactions with various ligands, for example immunoglobulins.

**Author Contributions:** Z.A., conceptualization, writing—original draft, supervision, review and editing; M.N.-R., data curation, investigation and writing—original draft, review and editing. Both authors have read and agreed to the published version of the manuscript.

**Funding:** This research received no external funding.

**Data Availability Statement:** Data was obtained from and are available ncadamcz@cyf-kr.edu.pl or ncnattic@cyf-kr.edu.pl.

**Acknowledgments:** This work was supported by the Statutory activity of the Jerzy Haber Institute of Catalysis and Surface Chemistry PAS.

**Conflicts of Interest:** The authors declare no conflict of interest.

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
