# Peer review of "Formation of Myoglobin Corona at Polymer Microparticles"

_colloids, doi:10.3390/colloids5020027_

Round 1

Reviewer 1 Report

The manuscript investigates the formation of a protein corona of myoglobin on the surface of polystyrene particles as a function of pH and ionic strength. The authors perform zeta potential, DLS and AFM measurements, thus giving results that are analysed in the light of the general electrokinetic model. This is a fundamental study in the field of colloid science, whose results may be of interest to the audience reached by the journal. Hence, I suggest its acceptance after some revisions, as stated below:

What was the specific reason for choosing polystyrene microparticles and the substrate for protein adsorption?

What is the amount/concentration of sulfate groups in the surface of polystyrene microparticles?

The equipment used for LDV measurements was not specified. Please, provide this information.

What was the reason for performing the adsorption experiments in NaCl solution and not in water? I suppose it was intended to simulate the ionic strength of body fluids, but this should be stressed out.

From Figure 1, pH is important to determine myoglobin stability over time, but, apparently, samples at pH 3.5 and 7.4 were not measured at the same time intervals, specially at longer times (points at 1200 and ca 1500 min). This should be explored in the new version of the manuscript. Are these data comparable? Moreover, results for [NaCl] = 0.1 M are not presented.

The negligible variation on the zeta potential of polystyrene particles with pH could be explained by in terms of pKa of sulfate groups on the surface of the particles. Since sulfonic acids are very strong (thus having very low pKa values), they may be deprotonated over the entire range of pH investigated. Please, consider this explanation.

It is not clear if the effective diffusion coefficient used in equation 1 was obtained by DLS. Please, specify this.

NaCl concentration is given in both molarity (M) and normality (N) in different parts of the manuscript. Please, standardize it.

Information that particles with protein corona at pH 7.4 form aggregates is important and should be used to better discuss the mechanisms of colloid stabilization involving the covered particles.

Number of replicates, for each experiment, needed to obtain the error bars displayed in Figs. 1, 3, 4, 5, 7.

English editing is recommended in the whole manuscript.

Author Response

Response to the points raised by the Reviewer

We are grateful for valuable remarks of the Reviewer, which has been considered in the revised version of our paper.

Reviewer 1

Comments and Suggestions for Authors:

The manuscript investigates the formation of a protein corona of myoglobin on the surface of polystyrene particles as a function of pH and ionic strength. The authors perform zeta potential, DLS and AFM measurements, thus giving results that are analysed in the light of the general electrokinetic model. This is a fundamental study in the field of colloid science, whose results may be of interest to the audience reached by the journal. Hence, I suggest its acceptance after some revisions, as stated below:

Comment 1: What was the specific reason for choosing polystyrene microparticles and the substrate for protein adsorption?

Reply 1: The polystyrene microparticles are characterized by low polydispersity, uniform and large negative charge density, which is additionally independent of pH. This creates a large stability of their suspension (latex) allowing efficient and irreversible adsorption of positively charged myoglobin molecules. 

Comment 2: What is the amount/concentration of sulfate groups in the surface of polystyrene microparticles?

Reply 2: This is an interesting remark. The number of sulfate groups can be most conveniently determined from the particle surface charge density, which can be calculated using the Gouy-Chapman relationship for a symmetric 1:1 electrolyte.

 σ0=[(8εkTnb)1/2/(0.160)]sinh[(eζ)/(2kT)]

where σ0 is the charge density expressed in the number of charges per one square nm, ε is the dielectric permittivity of water, k is the Boltzmann constant, T is the absolute temperature, nb is the number concentration of the salt (NaCl) expressed in m-3 and ζ is the zeta potential determined by the LDV electrophoretic mobility measurements. Using the data obtained in our work, i.e., = -85 and - 50 mV for 0.01 and 0.15 M NaCl one obtains σ0 = -0.18 and  -0.40 nm-2, respectively. This corresponds to one active sulfate group  per 5.6 or 2.5 nm2, respectively. It should be mentioned that in contrast to the titration method, these electrokinetic measurements yield  the uncompensated molecule charge and in consequence the physically significant number of active sulfate groups.

Comment 3: The equipment used for LDV measurements was not specified. Please, provide this information.

Reply 3: The electrophoretic mobility of myoglobin molecules and the particles were measured using the Laser Doppler Velocimetry (LDV) technique using the Zetasizer Nano ZS instrument (Malvern). This information was added to the revised version of the manuscript.

Comment 4: What was the reason for performing the adsorption experiments in NaCl solution and not in water? I suppose it was intended to simulate the ionic strength of body fluids, but this should be stressed out.

Reply 4: Indeed, in the blood serum the NaCl concentration is equal to 0.15 M. Howver, the main reason for performing the experiments in NaCl solutions was  that the experiments in pure water are irreproducible because of its low ionic strength below 10-5M and pH 5.6. For such low ionic strength and  this pH value, the adsorption of myoglobin on particles is  practically negligible.

Comment 5: From Figure 1, pH is important to determine myoglobin stability over time, but, apparently, samples at pH 3.5 and 7.4 were not measured at the same time intervals, specially at longer times (points at 1200 and ca 1500 min). This should be explored in the new version of the manuscript. Are these data comparable? Moreover, results for [NaCl] = 0.1 M are not presented. The negligible variation on the zeta potential of polystyrene particles with pH could be explained by in terms of pKa of sulfate groups on the surface of the particles. Since sulfonic acids are very strong (thus having very low pKa values), they may be deprotonated over the entire range of pH investigated. Please, consider this explanation.

Reply 5: These are valid remarks, which have been considered in the revised version of our manuscript. The time scale for Fig. 1 is now reduced to the time period of 400 minutes, which is much larger than the corona formation time of a few seconds. The results obtained for 0.15 M NaCl are added.

Indeed, the sulfonic acids are very strong hence, they remain fully ionized for the investigated pH range.

Comment 6: It is not clear if the effective diffusion coefficient used in equation 1 was obtained by DLS. Please, specify this.

Reply 6:  The relative diffusion coefficient of the myoglobin molecules relative to the molecules particles was calculated using the Smoluchowski approach as the sum of separate diffusion coefficients, which were experimentally determined by the DLS measurements.

Comment 7: NaCl concentration is given in both molarity (M) and normality (N) in different parts of the manuscript. Please, standardize it.

Reply 7: We only used molarity to express the NaCl concentration, the symbol N was used to express the surface concentration of myoglobin molecules adsorbed on the polymer particles.

Comment 8: Information that particles with protein corona at pH 7.4 form aggregates is important and should be used to better discuss the mechanisms of colloid stabilization involving the covered particles.

Reply 8: The aggregation occurs if the corona is formed at pH 7.4. If myoglobin is adsorbed at pH 3.5 the particles with the corona are stable for pH up to 9.

Comment 9: Number of replicates, for each experiment, needed to obtain the error bars displayed in Figs. 1, 3, 4, 5, 7.

Reply 9: The precision of the DLS and LDV measurements are given by the devices. However, in order to increase the accuracy of measurements they were repeated at least three times.

Comment 10: English editing is recommended in the whole manuscript.

Reply 10: This was done as suggested.

Reviewer 2 Report

The authors reported the kinetics of myoglobin adsorption on polystyrene particles in this manuscript. While the results are interesting, the writing and presentation of data need to be improved significantly. There are obvious grammar errors throughout the manuscript and some expressions are confusing. The formatting is also sloppy with inconsistent font sizes, random breaks in lines and disappeared labels in figures. The manuscript is hard to read and has to be proofread and polished meticulously before being considered to be published.

Line 17: Why the authors chose a pH of 3.5? Is this relevant to the physiological environment for myoglobin?

Line 54 – 75: Why this section had to be broken into multiple paragraphs with only one sentence in each paragraph? This is hard for readers to follow. Also the authors should summarize the effective information from reference in the background section instead of just pointing out what papers they cited.

Line 149 – 155: This is confusing – “assumed an average value”? What’s the NaCl concentration, 0.01 or 0.1 M? And after six hours it decreased to 4.2 from 1.2? Doesn’t make sense. The y-scale should also be modified for Figure 1, no resolution for data points < 400 min. The bottom left corner should be zoomed in.

Line 162 – 165: Is the diameter 830 or 810 at 0.01 M? The numbers are inconsistent. 10^(-2) should also be changed to 0.01 for consistence. In Figure 2, x label does not display properly.

Line 174: typo

Figure 3: y label does not display properly. Legends should also be added to ALL figures when apply.

Line 230: grammar error

Figure 4: Y label display issue.

Line281: Missing unit for NaCl.

Line 315: not readable.

Figure 5: AFM image should cover the y axis and y label does not display properly.

Figure 6: Y label issue. Also why the initial dH/dH0 value is smaller than 1? How were the values normalized?

Author Response

Response to the points raised by the Reviewer

We are grateful for valuable remarks of the Reviewer, which has been considered in the revised version of our paper.

Reviewer 2

Comments and Suggestions for Authors:

The authors reported the kinetics of myoglobin adsorption on polystyrene particles in this manuscript. While the results are interesting, the writing and presentation of data need to be improved significantly. There are obvious grammar errors throughout the manuscript and some expressions are confusing. The formatting is also sloppy with inconsistent font sizes, random breaks in lines and disappeared labels in figures. The manuscript is hard to read and has to be proofread and polished meticulously before being considered to be published.

These deficiencies, such as the inconsistent font sizes, random breaks in lines and disappeared labels in figures are caused by the Editor software formatting of our original text.

Comment 1: Line 17: Why the authors chose a pH of 3.5? Is this relevant to the physiological environment for myoglobin?

Reply 1: This is an interesting remark. Although pH in muscles, is well below 7.4 compared to the blood serum,  it is unlikely that it can drop to 3.5 under the stress conditions. Therefore, pH of 3.5 was selected not to mimic the physiological  behavior  but in order to optimize the corona formation procedure, which is the most efficient at pH below 4, where the molecules are  positively charged. If larger pHs are applied in the corona formation step, the maximum protein coverage decreases and the particles become less stable.

Comment 2: Line 54 – 75: Why this section had to be broken into multiple paragraphs with only one sentence in each paragraph? This is hard for readers to follow. Also the authors should summarize the effective information from reference in the background section instead of just pointing out what papers they cited.

Reply 2: This is a valid remark. We have accordingly corrected the introduction section and have added conclusions summing up the results obtained in the cited references.

Comment 3: Line 149 – 155: This is confusing – “assumed an average value”? What’s the NaCl concentration, 0.01 or 0.1 M? And after six hours it decreased to 4.2 from 1.2? Doesn’t make sense. The y-scale should also be modified for Figure 1, no resolution for data points < 400 min. The bottom left corner should be zoomed in.

Reply 3: We are thankful for these remarks. These sentences were corrected. The diffusion coefficient was measured for both 0. 01 and 0.15 M NaCl, the differences were insignificant yielding the average value of 1.2 +/- 0.1×10‐6 cm2 s‐1. Using the diffusion coefficient the molecule hydrodynamic diameter calculated to be 4.1 nm. It increased to 4.2 nm after six hours. This obvious mistake was corrected.

Comment 4: Line 162 – 165: Is the diameter 830 or 810 at 0.01 M? The numbers are inconsistent. 10^(-2) should also be changed to 0.01 for consistence. In Figure 2, x label does not display properly.

Reply 4: The bare particle diameter derived from DLS for 0.01 M NaCl was equal to 830 nm whereas that derived from laser diffractometry was equal to 810 nm but both agree within experimental error bounds. In further calculation the value of 830 nm was used. The label in Fig, 2 was increase in order to display properly.

Comment 5: Line 174: typo

Comment 6: Figure 3: y label does not display properly. Legends should also be added to ALL figures when apply.

Comment 7: Line 230: grammar error

Comment 8: Figure 4: Y label display issue.

Comment 9: Line281: Missing unit for NaCl.

Comment 10: Line 315: not readable.

Comment 11: Figure 5: AFM image should cover the y axis and y label does not display properly.

Reply 5-11: These are valid remarks, we have corrected these omissions and typos. However, the Figure legends were wrongly displayed during the text formatting by the editing software. 

Comment 12: Figure 6: Y label issue. Also why the initial dH/dH0 value is smaller than 1? How were the values normalized?

Reply 12: Indeed, this is a valid remark, the normalized initial dH/dH0  value should be equal unity. This figure has been accordingly  corrected.

Round 2

Reviewer 1 Report

The authors provided all requested information and made the appropriate changes. The manuscript can be published in the current form.

Reviewer 2 Report

The authors have addressed the comments. English editing is recommended.